# Effects of Cohousing Mice and Rats on Stress Levels, and the Attractiveness of Dyadic Social Interaction in C57BL/6J and CD1 Mice as Well as Sprague Dawley Rats

**DOI:** 10.3390/biology11020291

**Published:** 2022-02-11

**Authors:** Gerald Zernig, Hussein Ghareh, Helena Berchtold

**Affiliations:** Department of Pharmacology, Medical University of Innsbruck, Peter Mayr Strasse 1 a, 6020 Innsbruck, Austria; hussein.ghareh@i-med.ac.at (H.G.); helena.berchtold2@web.de (H.B.)

**Keywords:** cohousing, stress, CD1 mouse, C57BL/6J mouse, Sprague Dawley rat, fecal corticosterone and metabolites, cortisol, dyadic social interaction, conditioned place preference

## Abstract

**Simple Summary:**

Rats may kill mice. Therefore it is standard practice in many research animal housing facilities—despite often very limited space-to separate mice from rats (i.e., the predators) to minimize stress for the mice. We tested the effect of cohousing on the stress levels of mice from either the C57BL/6J (BL6) or the CD1 strain and Sprague Dawley rats by quantifying their fecal corticosterone and metabolites (FCM) concentration and investigated how cohousing impacts a behavioral assay, i.e., conditioned place preference for mouse-mouse or rat-rat social interaction. Mice from the BL6 strain (but not CD1 mice) that were cohoused with rats had significantly increased FCM concentrations, indicative of higher stress levels. In contrast to their elevated stress levels, the attractiveness for contextual cues associated with mouse-mouse social interaction even increased in rat-cohoused mice, albeit nonsignificantly. Thus, cohousing BL6 mice and rats did not impair a behavior of BL6 mice that had proved to be sensitive to social factors, especially handling by humans, in our laboratory. Our findings suggest that the effect of cohousing rats and mice on their stress levels and behavior might be less clearcut than generally assumed and might be overriden by conditions that cannot be controlled, i.e., different deliveries. Our findings can help to use research animal housing resources more efficiently.

**Abstract:**

Rats, including those of the Sprague Dawley strain, may kill mice. Because of this muricidal behavior, it is standard practice in many research animal housing facilities to separate mice from rats (i.e., the predators) to minimize stress for the mice. We tested the effect of cohousing on the stress levels of mice from either the C57BL/6J (BL6) or the CD1 strain and Sprague Dawley rats (SD rat) by quantifying their fecal corticosterone and metabolites (FCM) concentration. We also investigated cohousing impacts a behavioral assay, i.e., conditioned place preference for intragenus (i.e., mouse–mouse or rat–rat) dyadic social interaction (DSI CPP) that was shown be sensitive to social factors, especially to handling by humans. We found that the two delivery batches of BL6 mice or SD rats, respectively, had different stress levels at delivery that were statistically significant for the BL6 mice. Even so, the BL6 mice cohoused with rats had significantly increased FCM concentrations, indicative of higher stress levels, as compared to (1) BL6 mice housed alone or (2) BL6 mice at delivery. In contrast to their elevated stress levels, the attractiveness of contextual cues associated with mouse–mouse social interaction (DSI CPP) even increased in rat-cohoused BL6 mice, albeit non-significantly. Thus, cohousing BL6 mice and rats did not impair a behavioral assay in BL6 mice that was proven to be sensitive to handling stress by humans in our laboratory. SD rats cohoused with BL6- or CD1 mice, and CD1 mice cohoused with SD rats, showed DSI CPP that was not different from our previously published data on SD rats and BL6 mice of the Jackson- or NIH substrain obtained in the absence of cohousing. CD1 mice cohoused with rats did not show an increased FCM concentration compared to delivery. Our findings suggest that the effect of cohousing rats and mice under the conditions described above on their stress levels as opposed to their behavior might be less clearcut than generally assumed and might be overriden by conditions that cannot be controlled, i.e., different deliveries. Our findings can help to use research animal housing resources, which are usually limited, more efficiently.

## 1. Introduction

Rats [1], including those of the SD strain [2], may kill mice. Interestingly, far from all rats kill mice under animal behavioral laboratory experimental conditions: Bracy et al. 1978 found an overall killing rate by 60–75-day old male SD rats of only 28% [2]. These authors reported that the muricide rate in SD rats in their laboratory was only slightly above the killing rate reported previously [3,4,5]. Similarly, only about 20% of adult male Wistar rats investigated by Tulogdi et al. 2015 [1] killed mice with a 20 min cutoff time of the experiment (Haller, personal communication). In summary, upon closer inspection, muricide is not an obligatory rat behavior under controlled laboratory conditions.

Because of the perception of the rat as a predator (German term: “Fressfeind”, i.e., “devouring enemy”) of the mouse, it is standard practice in many research animal housing facilities to separate mice from rats to minimize stress for the mice. However, according to a limited informal survey by us, standard procedures may vary widely, both among commercial and academic breeders/experimental facilities, ranging from strictly separating mice and rats into different rooms throughout breeding and testing to cohousing mice and rats during breeding, albeit by using separate ventilation systems for each cage rack.

In many academic animal housing/testing facilities, space is a very limited resource that led, in our institution, to a de facto crowding out of behavioral research with rats in favor of mice, i.e., the genus with the larger pool of transgenic models. For the animal behavioral researcher studying social interaction, this is a harmful political/economic development, as rats are considered more ‘prosocial’ than mice, i.e., they show a more robust social behavior (see, e.g., [6,7,8,9]). On the other hand, mice should be protected as much as possible from stress during housing and testing, both for ethical and experimental design considerations.

For all of these reasons, we tested the hypothesis that mice experience more stress if cohoused with their likely predators, i.e., rats, by (1) quantifying stress levels through the concentration of fecal corticosterone and metabolites (FCM) [10,11,12,13] and by (2) performing a behavioral assay, i.e., conditioned place preference for intragenus (mouse–mouse or rat–rat) dyadic social interaction (DSI CPP [14]; for reviews see [15,16]), an assay that has been shown in rats to be very sensitive to social factors (i.e., greater size of the intragenus dyadic partner [17]) and, anecdotally, especially to handling by humans [18].

## 2. Material and Methods

### 2.1. Animals

Eight-week-old male Sprague Dawley rats (Crl:SD) or male mice of the C57BL/6J (JAX JAX^TM^) or CD1 strain (Crl:CD1(ICR)) were obtained from Charles River Laboratories (Sulzfeld, Germany; www.criver.com (accessed on 10 January 2022)), and transported by truck. At the Sulzfeld site of Charles River Laboratories, mice and rats are bred in the same rooms with each rack containing only one genus and with each rack being ventilated separately (personal communication). After intake in our laboratory, all animals were housed at a constant room temperature of 22 °C and had ad libitum access to tap water and pelleted chow from Ssniff Spezialdiaeten (Soest, Germany; www.ssniff.de (accessed on 10 January 2022)). Experiments were performed during the light phase of a continuous 12 h light/dark cycle with the lights on from 0800 h to 2000 h. Before the start of the CPP/CPA experiments, animals were singly housed for five to seven days and experienced a total of seven 2 min handling episodes with their allocated experimenter (at least one handling episode per day). The total mouse–rat cohousing vs. mouse–mouse intragenus housing period was slightly more than two weeks, i.e., 5–7 days of pre-experiment housing and 10 days of intra-experiment housing in single animal cages (totaling 15–17 days).

After the end of the CPP/CPA experiments, animals were euthanized with sevoflurane (Sevorane^®^) obtained from abbvie (Wien, Austria; www.abbvie.at (accessed on 10 January 2022)).

### 2.2. Conditioned Place Preference (CPP) for Dyadic Social Interaction (DSI)

The conditioned place preference for dyadic social interaction (DSI CPP) and for cocaine as performed in our laboratory was extensively validated and described [8,9,14,17,18,19,20,21,22,23]; for reviews, see [15,16]. Briefly, conditioning was conducted in a custom-made three-chamber CPP apparatus (64 cm wide × 32 cm deep × 31 cm high) made of unplasticized polyvinyl chloride. The middle (neutral) compartment (10 × 30 × 30 cm) had white walls and a white floor. Two doorways led to the two conditioning compartments (25 × 30 × 30 cm each) with walls showing either vertical or horizontal black-and-white stripes of the same overall brightness [16] and stainless steel floors containing either 168 holes (diameter 0.5 cm) or 56 slits (4.2 × 0.2 cm each). A systematic investigation of the time spent in each conditioning compartment in a pretest session did not reveal any compartment bias (i.e., we used a nonbiased apparatus; data not shown). Time spent in each compartment was digitally recorded with a video camera and analyzed offline with hand timers. The CPP apparatus was cleaned with a 70% camphorated ethanol solution after each session. All experiments were performed under neon ceiling light (58 W, 1 m distance) and white noise from continuously running allergen filter boxes. Of note, all experiments were performed by the same experimenter (HB). Figure 1 shows the CPP apparatus and two C57BL/6J mice during a conditioning session.

Our conditioning procedure was previously described and discussed in detail [14,15,16,18,24]. For the acquisition of CPP for DSI, the conditioning procedure comprised a pretest session on day 1, followed by eight consecutive training days in an alternate-day design of the pattern DSI-sal-DSI-sal-DSI-sal-DSI-sal (one training session per day; for a schematic representation, see Figure 1 of [14]). CPP was tested on day 10. In the DSI group, the stimuli were either (1) a 15 min dyadic social interaction session with a sex- and weight-matched male conspecific preceded by an intraperitoneal (i.p.) injection of 10 mL/kg saline, or (2) only a saline injection as the comparator stimulus. Pretest bias for any of the two conditioning chambers was declared if, during pretest the animal spent more time in one of the conditioning chambers. The initially non-preferred chamber was subsequently paired with the stimulus of interest (noncounterbalanced compartment allocation; see [15,16] for a detailed discussion).

### 2.3. Hierarchy Analysis: Scoring of Dominance vs. Subordination

The last of the four DSI episodes during CPP training was videorecorded and evaluated offline for signs of dominance/subordination in each mouse pair strictly according to the scoring system by Bakker and colleagues [25] and as previously described [18]: Aggressive dominance (a hierarchy score of h3) was defined as three consecutive attacks by one mouse (aggressive grooming, biting and chasing); passive dominance (a score of h2) was defined as consistent threatening displacement by one mouse, including upright or sideways postures; subordinate behavior (score of h0) was defined as retreat or fleeing by one mouse including “on back” position and crouching, and a draw (a score of h1) was defined as no attacks or consistent displacement occurring on the part of either mouse. Although the scoring experimenter was instructed to ignore all previously collected information on the individual mice, the offline hierarchy analysis was performed by the same experimenter who had previously quantified the time spent by the respective mice in the subsequent CPP test, so blinding to the behavior in the subsequent CPP was not absolute. However, due to the large number of video recordings, actual blinding seems plausible in most of the cases.

### 2.4. Fecal Corticosterone and Metabolites (FCM) Assay

Each fecal sample was analyzed in duplicate using a corticosterone (competitive) enzyme-linked immunosorbent assay (ELISA) kit EIA-4164 from DRG Instruments GmbH (Marburg, Germany; www.drg-diagnostics.de, accessed on 10 January 2022). The diagnostic kit was originally produced to analyze corticosterone in human samples. However, because wells are coated with polyclonal anti-corticosterone antibody (polyclonal antibody from rabbit), the kit can be used to quantify FCM in rodents as well [26].

All the fecal samples were collected from groups at various time points, i.e., at the time of delivery (between 0800 h and 1200 h), and immediately after the CPP test (between 1000 h and 1800 h). The groups were sorted based on their housing conditions. Fecal boli were stored at −80 °C until quantification. Corticosterone was shown to be a stable molecule, and corticosterone levels change less than 10% even when they are stored at room temperature for 24 h [27].

Fecal boli were thawed, weighted, and submerged in 96% (*v*/*v*) ethanol. Next, we added 3 mL of ethanol 96% for 1 g of feces. All samples were vortexed vigorously and incubated on a shaking device overnight. On day 2, samples were centrifuged at 15,000 rpm for 20 min. A 1.5 mL aliquot of the supernatant was collected carefully and centrifuged at 15,000 rpm for a further 10 min. A volume of 200 μL of the supernatant was diluted in ethanol (final dilutions of 1:2 to 1:10 were used) and analyzed.

### 2.5. Statistical Methods

Group statistics (i.e., mean standard error of mean (SEM)), correlation coefficients and *t*-tests (1- or 2-sided, homo-or heteroskedastic as appropriate) were calculated using Microsoft^®^ Excel for Mac^®^ (version 15.29.1) and Prism^®^ 7.0 (www.graphpad.com, accessed on 10 January 2022).

## 3. Results

### 3.1. Stress Levels as Quantified by FCM

Table 1 shows the stress levels—as quantified by FCM concentrations—in the different experimental groups at delivery and after the CPP test and gives p values for the different across-group comparisons. Of note, all FCM concentrations were quantified after the behavioral experiments had been completed by an experimenter (HG) who had not performed the DSI CPP experiments and was de facto blind to the behavioral treatments.

Housing BL6 mice alone did not change their stress levels between delivery and the CPP test 15–17 days later. Stress levels determined after the CPP test (Table 1) significantly increased in BL6 mice when cohoused with SD rats compared to (1) their FCM concentrations at delivery and (2) the post-CPP-test FCM concentrations of mice that had not been cohoused with rats. However, the two different BL6 mouse batches also significantly differed from each other at delivery, with a low FCM concentration at delivery of the BL6 mice that were later to be cohoused with SD rats (Table 1). Therefore, differences between groups may have been exaggerated. However, the statistical significance remained high when comparing the FCM concentration of BL6 mice cohoused with rats with the FCM concentration of the pooled BL6 mice at delivery (Table 1). To conclude, the increase in FCM concentration as a measure of stress increased in the BL6 mice that were cohoused with rats for 15–17 days.

### 3.2. Behavior

Conditioned place preference for contextual stimuli associated with intragenus (i.e., mouse-mouse or rat-rat) dyadic social interaction was even increased, albeit non-significantly, in BL6 mice cohoused with rats as compared to BL6 mice housed alone (Table 2). As shown previously, SD rats showed a more robust DSI CPP than the mice, a genus considered less prosocial than rats (see Introduction section). Similar to BL6 mice, CD1 mice cohoused with rats also showed robust DSI CPP (Table 2).

At the level of the individual animal (Table 3), stress (FCM) levels were correlated only poorly and non-systematically with DSI CPP (i.e., time spent in the DSI-associated compartment minus time spent in the saline-associated compartment).

We also determined the hierarchic position of the two animals at the last of four pairings and tried to correlate the hierarchy score with the degree of DSI CPP. No relevant correlation was found for any of the groups (data not shown). Finally, we tried to quantify stress levels by measuring the fecal output of the animals [28]. This, however, proved not to be feasible within a reasonable time frame for feces collection.

## 4. Discussion

Our findings with BL6 mice and SD rats confirm the general notion that cohousing mice with rats, i.e., their likely predators, increases the stress levels of the mice as quantified by the concentration of fecal corticosterone and metabolites (FCM; Table 1). In contrast, the effect of cohousing BL6 mice and SD rats on a behavioral assay that is sensitive to social factors [17,18], and especially sensitive to stress induced by handling by humans ([18] and Zernig, unpublished observation), i.e., conditioned place preference for intragenus (i.e., mouse–mouse or rat–rat) dyadic social interaction, were surprising: Contrary to what many in the field may argue, cohousing did not impair this stress-sensitive behavioral assay in any of the tested animal strain or species, i.e., mice of the BL6 or the CD1 strain or rats of the Sprague Dawley strain (Table 2). In addition, when studying group sizes (*n* = 8) that are generally considered sufficient by animal experimental review boards, we found that stress levels differed between delivery batches of mice and Sprague Dawley rats. Of note, all behavioral experiments were performed by the same experimenter (HB) to exclude an experimenter effect [18].

Our findings on fecal corticosterone concentrations differ from those on plasma corticosterone concentrations by Greene and coworkers, who demonstrated an overall increase from day 0 to 15 both for C57BL/6NCrl mice that were housed in a separate room and for mice that were cohoused with rats in the same room and subjected to their olfactory or visual stimuli or a combination thereof [29], with no systematic differences between the individual groups (their Figure 4). We argue that our sampling of fecal pellets would have a lesser impact on the parameter under investigation, i.e., stress hormone concentration, than taking submandibular blood [29]. Additionally, because of the considerable intestinal transit time (estimates vary between 8 and 12 h; see, e.g., [6]), fecal corticosterone can be thought of as a more integral stress measure than plasma corticosterone. Finally, C57BL/6 substrains (i.e., from the Jackson or Charles River Laboratories) have different behavioral profiles [18,30,31].

Interestingly, at the group level, increased stress (FCM) levels in BL6 mice were associated with an (albeit statistically nonsignificant) increase in DSI CPP, as if higher stress levels due to the presence of a predator caused mouse–mouse social interactions to become more attractive for the mice, the mouse genus being notoriously poor in prosocial behavior as compared to rats (see, e.g., [7,8,9]). At the individual animal level, correlation between stress (FCM) levels and the attractiveness of DSI was generally poor and nonsystematic (Table 3). As shown previously for mice [18], there was no correlation between the hierarchic position of the animal in the last pairing session and the degree of DSI CPP, again due to the fact that, in the overwhelming majority of the cases, no hierarchy developed during the four pairings of the conditioning procedure as previously demonstrated [18].

Confirming previous findings of our group [8], Sprague Dawley rats found contextual stimuli associated with dyadic social interaction more attractive than mice (Table 2). The fact that rats generally had FCM concentrations that were roughly one order of magnitude higher than mice corroborates previous findings by others (see, e.g., [12]).

SD rats cohoused with BL6- or CD1 mice and CD1 mice cohoused with SD rats showed DSI CPP that was not different from our previously published data on SD rats and BL6 mice of the Jackson- or NIH substrain obtained in the absence of cohousing, i.e., after intragenus housing only (see [15,16] for reviews; [18] for BL6 substrain differences).

The limitations of our investigation are, first of all, the limited number of experimental groups and group sizes. Our hands were tied by the nature of our investigation: We had proposed to test a widely held tenet of experimental animal housing, i.e., that the cohousing of mice and rats severely impacts the behavior of the mice. Regulatory bodies required us to limit the number of animals per group to eight and the number of experimental groups to the absolute minimum to prove or disprove the tenet.

Another limitation of our study is the specificity of the experimental conditions in our laboratory and the caveat that our findings may, therefore, not be generalizable. Animals (mice and mice or mice and rats) were singly housed in adjacent de facto semitransparent cages that shared the same ventilation system (i.e., cages on shelves with the air sucked through a barrier and around the single cages to an outlet at the top of the shelves) for a total of only slightly more than two weeks. The CPP test apparatus was located beyond the ventilation/allergy barrier, again behind a de facto semitransparent hard curtain with ventilation holes in it.

Finally, the behavioral test used, DSI CPP, may be insensitive to stress. This is unlikely since previous work by our group demonstrated a distinct experimenter effect (i.e., handling by a human) in BL6 mice [18]. Accordingly, great care is taken in our lab to handle the animals often before the start of the behavioral experiment (see Methods section). SD rats were also found to be sensitive to the stress of handling by humans in our laboratory [18], in some cases completely disrupting subsequent DSI CPP (Zernig, unpublished observation).

## 5. Conclusions

We found that the effect of cohousing mice and rats did not impair a behavioral assay that is sensitive to social factors and very sensitive to handling stress by humans, although cohousing increased stress (FCM) levels in BL6 mice at group sizes of *n* = 8. Furthermore, different delivery batches of C57 mice and SD rats had different stress levels at delivery. Our findings suggest that the effect of cohousing rats and mice under the conditions described above on their stress levels and behavior might be less clearcut than generally assumed and might be overridden by conditions that cannot be controlled, i.e., at different deliveries. With respect to the “refine” component of the “3R” guidelines for animal experiments, our findings show that cohousing significantly increases FCM concentrations, indicative of increased stress, which is not correlated by an impairment in a behavioral experiment (DSI CPP) shown to be very sensitive to the effect of handling by humans. In line with previous experiments by other groups ([29,32]; but see, e.g., [33]), our findings suggest that it may not be absolutely necessary to separate mice from rats during the performance of behavioral experiments, thus optimizing the use of often very limited animal housing resources. Future experiments with larger group sizes performed in different laboratories could corroborate or refute the robustness of our findings.

## Figures and Tables

**Figure 1 biology-11-00291-f001:**
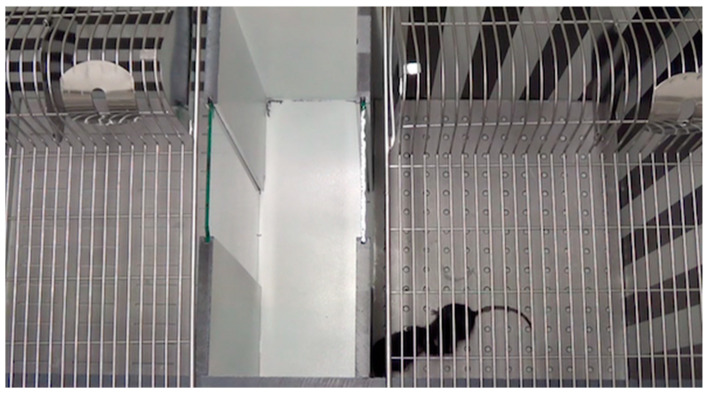
Social interaction between two male C57BL/6J mice during conditioned place preference (CPP) training. Shown is a single frame of a video recording of a conditioning session. The social interaction partners are confined to the conditioning compartment (all sliding doors between the middle/neutral chamber and the conditioning compartments closed). To prevent the mice from jumping out of the CPP apparatus, regular wire mesh cage tops are put on top of the conditioning compartments (spout guard and inverted pellet trough visible). The different floors and wallpaper patterns of the two conditioning compartments can be clearly distinguished.

**Table 1 biology-11-00291-t001:** Stress levels quantified by FCM concentrations at delivery and after the mouse–mouse DSI CPP test. *t* tests were 2-sided and either homo- or heteroskedastic as appropriate. Shown are FCM concentrations in nmol/l: na, not available; nj, not justified statistically.

Experimental Group (Group Size)	FCM at Delivery (nmol/L; Mean ± SEM)	FCM after CPP Test (nmol/L; Mean ± SEM)
Mouse BL6 alone (*n* = 8)	50 ± 7	31 ± 8 (*p* = 0.11 compared to delivery) (*p* = 0.26 homoskedastic compared to pooled BL6 at delivery)
Mouse BL6 cohoused with rat SD (*n* = 8)	33 ± 4 (*p* = 0.047 homoskedastic compared to Bl6/j alone)	74 ± 8 (*p* = 0.0025 homoskedastic compared to BL6 alone) (*p* = 0.0005 homoskedastic compared to delivery) (*p* = 0.0008 homoskedastic compared to pooled mouse BL6 at delivery)
Mouse BL6 pooled (*n* = 16)	41 ± 4	nj
Mouse CD1 alone	na	na
Mouse CD1 cohoused with rat SD (*n* = 8)	49 ± 11	54 ± 11 (*p* = 0.76 homoskedastic compared to delivery)
Rat SD alone	na	na
Rat SD cohoused with mouse Bl6/J (*n* = 8)	285 ± 83	479 ± 197 (*p* = 0.38 heteroskedastic compared to delivery)
Rat SD cohoused with mouse CD1 (*n* = 8)	475 ± 89 (*p* = 0.14 homoskedastic compared to rats cohoused with Bl6j)	342 ± 50 (*p* = 0.51 heteroskedastic compared to rats cohoused with BL6) (*p* = 0.22 homoskedastic compared to delivery)
Rat SD pooled (*n* = 16)	380 ± 64	410 ± 100 (*p* = 0.80 homoskedastic compared to delivery)

**Table 2 biology-11-00291-t002:** Conditioned place preference for dyadic social interaction in mice housed alone or cohoused with rats, and in rats. Of note, the dyadic social interaction was always intragenus, i.e., mouse–mouse or rat–rat. The table shows the times (in seconds, means ± SEM; group size was always 8 animals) spent in the compartment previously associated with dyadic social interaction following an i.p. saline injection (DSI) or saline injection alone (sal). Neu, a neutral compartment located between the conditioning compartments. Time spent in the DSI compartment was statistically compared to time spent in the sal compartment within each group for each animal assuming a CPP for DSI (i.e., one-sided unpaired *t*-test). Across-group statistical comparisons for DSI-sal were performed with a two-sided unpaired *t*-test. For better transparency, DSI-sal is shown here as the difference between the rounded mean values. For statistical comparisons, the DSI-sal difference was calculated for each individual animal, thus leading to a mean rounded DSI-sal of 56 s (vs. 55 s) for the BL6 group and of 192 s (vs. 191 s) for the rat cohoused with the mouse CD1 group.

Experimental Group	Time Spent in DSI Compartment (s)	Time Spent in Neutral Compartment (s)	Time Spent in Sal Compartment (s) (*p* Compared to DSI Compartment)	DSI-Sal (s)
Mouse BL6 alone	321 ± 37	313 ± 14	266 ± 37 (*p* = 0.24)	55
Mouse BL6 cohoused with rat SD	346 ± 31	320 ± 28	234 ± 18 (*p* = 0.017)	112 (*p* = 0.52 compared to BL6 alone)
Mouse CD1 alone	na	na	na	na
Mouse CD1 cohoused with rat SD	392 ± 24	251 ± 24	258 ± 23 (*p* = 0.0059)	134 (*p* = 0.71 compared to BL6 cohoused with rat)
Rat SD alone	na	na	na	na
Rat SD cohoused with mouse BL6	362 ± 27	288 ± 35	251 ± 35 (*p* = 0.034)	111
Rat SD cohoused with mouse CD1	428 ± 31	235 ± 18	237 ± 31 (*p* = 0.0074)	191 (*p* = 0.33 compared to rat cohoused with BL6)

**Table 3 biology-11-00291-t003:** Correlation between stress levels quantified by FCM and intragenus (i.e., mouse–mouse or rat–rat) dyadic social interaction as a behavioral measure of stress. na, not available.

Experimental Group	Correlation between FCM at Delivery and DSI CPP	Correlation between FCM after CPP Test and DSI CPP
Mouse BL6 alone (*n* = 8)	−0.47	0.17
Mouse BL6 cohoused with rat SD (*n* = 8)	−0.29	0.04
Mouse BL6 pooled (*n* = 16)	-0.45	0.21
Mouse CD1 alone	na	na
Mouse CD1 cohoused with rat SD (*n* = 8)	0.63	0.29
Rat SD alone	na	na
Rat SD cohoused with mouse BL6 (*n* = 8)	−0.28	0.36
Rat SD cohoused with mouse CD1 (*n* = 8)	0.76	0.73
Rat SD pooled (*n* = 16)	0.36	0.30

## Data Availability

The authors confirm that the data supporting the findings of this study are available within the article.

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
