# Peer review of "Effects of Cohousing Mice and Rats on Stress Levels, and the Attractiveness of Dyadic Social Interaction in C57BL/6J and CD1 Mice as Well as Sprague Dawley Rats"

_biology, 2022, doi:10.3390/biology11020291_

Round 1
Reviewer 1 Report
Dear Authors,
I carefully read the submitted manuscript. All performed experiments are conclusive and the paper is well written. Therefore, for me the paper is accepted in present form.
All the best
1. What is the main question addressed by this research?
The main question is, if cohousing of mice and rats (sometimes necessary, because of limited space) has an influence on the stress level of the animals (yes) and if it´s possible to use the cohoused animals for a behavioral assay without a negative influence on the results.
2. Consideration if the topic is original and relevant in the field?
Due to my knowledge, there is no publication, which analyzed the cohousing effects on behavioral assays. The topic is relevant, because this results may facilitate the efficiency of housing capacity’s, especially in waiting areas for behavioral assays.
3. What does it add to the subject area compared with other published material?
In a publication from Thomas M Greene they analyzed whether visual, olfactory, or combined stimuli are responsible for stress-related changes in mouse physiology and behavior when cohoused with rats, not in one cage but in one room.
In the present paper, they analyzed “a real cohousing” which might be necessary cause of limited space in some areas.
4. What specific improvement could the authors consider regarding the methodology?
There could be an improvement with the control groups, but in my opinion it´s always possible to have more or better control groups. The animal number was low and the robustness of the findings might be higher with a larger animal number. However, in Germany a power analysis is necessary for the experimental proposal, so maybe they could include a power analysis to undermine the findings with the used number of animals.
5. Are the conclusions consistent with the evidence and arguments presented and do they address the main question posed?
Yes
6. Are the references appropriate?
Yes (of course it´s always possible to use more)
7. Comments on the tables and figures
For me the tables and figures are understandable. It would be great to have a picture of the used chamber for the behavioral experiments to make easier to understand the time the animals spent in the different compartments.
Author Response
You wrote:
Dear Authors,
I carefully read the submitted manuscript. All performed experiments are conclusive and the paper is well written. Therefore, for me the paper is accepted in present form.
Our response: Thank you very much for your kind words. We would like to express our gratitude for your time and your helpful comments.
You wrote: 3. What does it add to the subject area compared with other published material?
In a publication from Thomas M Greene they analyzed whether visual, olfactory, or combined stimuli are responsible for stress-related changes in mouse physiology and behavior when cohoused with rats, not in one cage but in one room.
In the present paper, they analyzed “a real cohousing” which might be necessary cause of limited space in some areas.
Our response: Thank you very much for alerting us to the work by Greene et al 2014. We have mentioned and discussed this article in the revised manuscript.
You wrote: 4. What specific improvement could the authors consider regarding the methodology?
There could be an improvement with the control groups, but in my opinion it´s always possible to have more or better control groups. The animal number was low and the robustness of the findings might be higher with a larger animal number. However, in Germany a power analysis is necessary for the experimental proposal, so maybe they could include a power analysis to undermine the findings with the used number of animals.
Our response: With respect to your comments regarding the group size of 8 animals, we beg to disagree. As already hinted at in the discussion of our original manuscript (lines 213f) and as already explicitly stated later in the discussion (lines 239f), we were warned by animal experimental review boards NOT to exceed a group size of 8 animals which was considered “very generous” by the regulatory bodies. In fact, we had previously published statistically sound social interaction CPP data with as little as 5 animals per group (see, e.g., figure 1 of Fritz et al 2011, also mentioned in our response to reviewer 2). Based on the successful demonstration of the statistical significant differences in a variety of measures, the animal experimental review boards in fact directed us not to exceed a group size of 8 animals in the present experiments.
In Austria, a power analysis is not required for animal research proposals.
In support of our argument, in the article by Greene et al 2014 that you had generously alerted us to in your item 3 (above), figure 4 (the plasma corticosterone concentrations) gives the group sizes: n=8 each.
You wrote: 7. Comments on the tables and figures
For me the tables and figures are understandable. It would be great to have a picture of the used chamber for the behavioral experiments to make easier to understand the time the animals spent in the different compartments.
Our response: Following your suggestion, we have added a picture (a frame of a video) showing social interaction of two C57BL/6J mice with all the salient features of the CPP apparatus described.
Reviewer 2 Report
The authors assessed the effects of cohousing mice and rats on stress levels, measured by non-invasive faecal assessments, and on social interaction by conditioned place preference for intra-genus dyadic social interaction. The results found sound interesting but, as stated even by the authors, the low number of animals tested could have affected the relevance and the replication of the data. I understand that they were forced to reduce the tested animals, but 8 animals per group are the minimum for statistical analysis, especially when behavioural experiments are performed.
Abstract
Line 13 In rodents the principal glucocorticoid hormone is corticosterone, thus, in their faeces, you will find basically corticosterone metabolites.
Introduction
Lines 37-40 The details of Bracy's results don’t explain anything more about the rat muricide rate thus can be removed to increase readability. Please add references about previous data about killing rate.
Line 42 Why did you cite a personal communication when it is possible to find reviewed published data??
Lines 66-69 I suggest moving this sentence in the material and methods section
Material and methods section
Line 78 correct "ad libitrum" with "ad libitum"
Line 103 A graphical scheme could be useful to clarify the experimental design.
When did you perform the faecal collection to assess FCM levels at delivery? It is well known that a relocation, but also a novel environment, activates the HPA axis and consequently increases the secretion of glucocorticoids. However, not all the animals show a similar response to a stimulus thus this could be the reason for the difference that you found in FCM levels among different groups. Would have been better if you have had animals in basal condition before starting your experimental design.
Did you consider the animal hormonal circadian rhythm excretion pattern when you performed the faecal collection for FCM at delivery and after the CPP test? Considering the intestinal transit time in rodents (estimated in 8-12 hours) it is important to know when you performed the collection.
Results
In two groups of animals, Rat alone and mouse CD1 alone, you didn’t assess FCM levels, which is the reason? These data could have explained the stress level of each animal group after the behavioural experience in comparison with their basal levels.
Three tables and zero graphs. If you remove some data and convert it into 1-2 graphs the readability of the manuscript will increase a lot.
Discussion line 206 same problem as in the abstract, please correct it.
Author Response
Thank you very much for your time and your helpful comments.
You wrote: I understand that they were forced to reduce the tested animals, but 8 animals per group are the minimum for statistical analysis, especially when behavioural experiments are performed.
Our response: We beg to disagree. As already hinted at in the discussion of our original manuscript (lines 213f) and as already explicitly stated later in the discussion (lines 239f), we were warned by animal experimental review boards NOT to exceed a group size of 8 animals which was considered “very generous” by the regulatory bodies. In fact, we had previously published statistically sound social interaction CPP data with as little as 5 animals per group (see, e.g., figure 1 of Fritz et al 2011, detailed again below).
You wrote:
Abstract
Line 13 In rodents the principal glucocorticoid hormone is corticosterone, thus, in their faeces, you will find basically corticosterone metabolites.
Our response: Text changed according to your suggestion.
You wrote: Introduction
Lines 37-40 The details of Bracy's results don’t explain anything more about the rat muricide rate thus can be removed to increase readability. Please add references about previous data about killing rate.
Our response: Text shortened and reference given according to your suggestion.
You wrote: Line 42 Why did you cite a personal communication when it is possible to find reviewed published data??
Our response: In fact, the killing rate was NOT reported in the paper by Tulogdi et al 2015 itself. We had to ask one of the coauthors – hence the “personal communication”. Following your suggestion, however, we have included the paper itself in the same sentence.
You wrote: Lines 66-69 I suggest moving this sentence in the material and methods section.
Our response: Following your suggestion, this sentence has been moved to the end of section 2.1.
You wrote:
Material and methods section
Line 78 correct "ad libitrum" with "ad libitum" .
Our response: The text had no typo “libitRum”. Did you mean we should use italics for “ad libitium”? We have changed the text accordingly.
You wrote: Line 103 A graphical scheme could be useful to clarify the experimental design.
Our response: We beg to disagree: The schedule is so simple that we would opine that the text is descriptive enough. However, following your suggestion we have directed the interested reader to a schematic diagram in a previous publication, i.e., Fritz et al 2011.
You wrote: When did you perform the faecal collection to assess FCM levels at delivery? It is well known that a relocation, but also a novel environment, activates the HPA axis and consequently increases the secretion of glucocorticoids. However, not all the animals show a similar response to a stimulus thus this could be the reason for the difference that you found in FCM levels among different groups. Would have been better if you have had animals in basal condition before starting your experimental design.
Our response: Because of limited resources, many experimenters would, according to our experience, start rodent experiments very soon after delivery of the animals. Also, we wanted to explicitly investigate what the impact of the delivery process on the animals’ stress levels were. Your suggestion is very valuable: It would have been great to systematically investigate which time span the different rodent genus and strains need to reach basal stress levels after delivery. Testing this in a meaningful systematic way, however, was beyond the scope of the present study.
You wrote: Did you consider the animal hormonal circadian rhythm excretion pattern when you performed the faecal collection for FCM at delivery and after the CPP test? Considering the intestinal transit time in rodents (estimated in 8-12 hours) it is important to know when you performed the collection.
Our response: As already stated in lines 137f of the original manuscript, the fecal samples were collected at various time points. Following your suggestion we have now explicitly stated the respective times of the day.
You wrote:
Results
In two groups of animals, Rat alone and mouse CD1 alone, you didn’t assess FCM levels, which is the reason? These data could have explained the stress level of each animal group after the behavioural experience in comparison with their basal levels.
The reason for this was scarcity of experimenter time and resources. As the primary focus of the study was and is the stress levels vs the behavior of the C57BL/6J mice, we would ask the reviewer to forgive us this shortcoming.
Three tables and zero graphs. If you remove some data and convert it into 1-2 graphs the readability of the manuscript will increase a lot.
Our response: With respect to converting table 1 or table 2 to a figure, we would like to keep the table format as the wealth of data would make their presentation in figure form unwiedly and confusing for the reader. Presenting the data shown in table 3 in figure form would, in our opinion, be nonsensical. Following the suggestion of reviewer 2, we have now added a figure showing two C57BL/6 mice in social interaction during a CPP conditioning session.
You wrote: Discussion line 206 same problem as in the abstract, please correct it.
Our response: Text changed according to your suggestion.
Round 2
Reviewer 2 Report
The authors improved the manuscript and the revised version is acceptable in its current form.